# Polycyclic Aromatic Hydrocarbons (PAHs) Occurrence in Traditionally Smoked Chicken, Turkey and Duck Meat

Cristian Ovidiu Coroian [1], Aurelia Coroian [2,*], Anca Becze [3], Adina Longodor [4], Oana Mastan [5] and Răzvan-Mihail Radu-Rusu [6,*]

[1] Department of Animal Nutrition and Feeding, Faculty of Animal Sciences and Biotechnology, University of Agricultural Sciences and Veterinary Medicine Cluj-Napoca, Calea Mănăștur 3–5, 400372 Cluj-Napoca, Romania

[2] Department of Toxicology, Faculty of Animal Sciences and Biotechnology, University of Agricultural Sciences and Veterinary Medicine Cluj-Napoca, Calea Mănăștur 3–5, 400372 Cluj-Napoca, Romania

[3] Research Institute for Analytical Instrumentation, National Institute of Research and Development for Optoelectronics INOE 2000, 67 Donath Street, 400293 Cluj-Napoca, Romania

[4] Department of Land Measurements and Exact Sciences, Faculty of Forestry and Cadastre, University of Agricultural Sciences and Veterinary Medicine Cluj-Napoca, Calea Mănăștur 3–5, 400372 Cluj-Napoca, Romania

[5] Department of Anatomy, Faculty of Veterinary Medicine, University of Agricultural Sciences and Veterinary Medicine Cluj-Napoca, Calea Mănăștur 3–5, 400372 Cluj-Napoca, Romania

[6] Department of Animal Resources and Technology, Faculty of Food and Animal Sciences, Ion Ionescu de la Brad University of Life Sciences, 8 Mihail Sadoveanu Alley, 700489 Iasi, Romania

* Correspondence: aurelia.coroian@usamvcluj.ro (A.C.); radurazvan@uaiasi.ro (R.-M.R.-R.)

**Abstract:** An increasingly high interest is given to the sensory, nutritional, and sanogenic qualities of meat. Considering that poultry meat is nowadays the main quantitatively demanded meat for human consumption, its quality is largely verified and monitored. Toxic compounds are trace markers to be monitored, as their health impacts often cause a high health risk for humans. We have evaluated how a traditional method of meat preservation—hot smoking with natural wood smoke—adds certain polycyclic aromatic hydrocarbons (PAHs) to chicken, duck, and turkey meat. One- vs two-day smoking period and three wood types for smoking (plum, cherry, and beech) have shown that the highest concentrations of PAHs were present in duck meat, irrespective of smoking time or wood type. A higher concentration overall of PAHs was quantified when beech wood was used, followed by cherry and plum woods. Fluorene associated with beech wood gave the highest values for day 1 and day 2, followed by duck and turkey meat, respectively. Very significant differences ($p < 0.001$) were usually observed for duck meat when compared with chicken and turkey meat, but it was also easy to notice absolute values for Anthracene, Phenanthrene, or Fluoranthene. As expected, two-day smoking contributed to higher concentrations of PAHs in meat.

**Keywords:** polycyclic aromatic hydrocarbons; meat; chicken; duck; turkey

## 1. Introduction

Meat is one of the main protein sources in the human diet and an important component of a balanced diet [1]. Short-generation interval animal species, such as poultry and pig, provide the largest amount of meat globally. The previsions indicate these species as the main sources of meat for the upcoming period, poultry being among consumers' top preferences [2]. Worldwide, poultry is the second top meat produced as primary livestock and yields after pork, while turkey reaches 5–6% of chicken trades and duck meat is slightly above 4% [2]. While chicken meat production increased by about 50% throughout the past decade, turkey has increased slightly by about 8–9% and duck by about 40%. However, market demands for duck meat are far from being met, which is the same for turkey and even for chicken [2]. Duck requires the most particular rearing conditions and, therefore,

has regional specificity. Both turkey and duck meats have different nutritional and sensory qualities as compared with chicken, and global demands have an ascending trend.

Besides production aspects, meat quality is an increasingly imperative requirement, especially among rich countries' consumers. The nutritional values seem to reach a stable level based on the applied farming technologies. Meat quality is a complex of features, of which the conditions of being safe, sanogenic, and from a trustworthy source have probably become the most important. Among multiple preserving and preparation methods, meat smoking imprints special sensory characteristics [3], rendering it very attractive for consumers. Such preservation was most likely begun in prehistoric times [4] and still represents a practice elsewhere. Smoking temperature may give more desirable aroma and flavour to meat [5]. Despite such benefits, wood smoking also brings into meat polycyclic aromatic hydrocarbons (PAHs) and their alkylated derivatives [6]. PAHs are carcinogenic, and thermal cooking processes add them to various foods [7]. Cooking and processing techniques, such as roasting, barbecuing, grilling, smoking, heating, drying, baking, ohmic-infrared cooking, etc., also contribute to PAHs formation. Various factors, such as the distance from the heat source, used fuel, level of processing, and cooking durations and methods, as well as processes, such as reuse, conching, concentration, crushing, and storage, all increase the concentration of PAHs in food [8]. Different cooking methods also contribute to the formation of heterocyclic aromatic amines (HCAs)—another mutagenic and/or carcinogenic compound naturally formed during the cooking of protein-rich foods, such as beef, poultry, and fish—which exhibit higher accumulation when the deep-fat frying method is used in food processing [9].

Specific legislation [10] limits the concentration of PAHs in foods. A wide-ranging debate is in full swing and will probably continue with the emergence of new data. This is due to associated toxicity and health issues that PAHs have been linked to [11]. Since the 1970s, a set of 16 PAHs have been monitored by the United States Environmental Protection Agency [12], while the European Food Safety Agency (EFSA) identified $\Sigma$8PAHs or $\Sigma$4PAHs as relevant toxicity indicators [12,13].

Not all types of wood produce the same amount of PAHs; large variation limits are described [14,15]. The time of smoking also dramatically contributes to final PAH concentrations in meat [13], as do the various smoke generating methods [16,17]. Although PAHs are naturally occurring in the environment, including in the air we breathe [18–20], the most acute toxic effect is given by direct dietary ingestion [21]. Of the many adverse effects of PAHs, those leading to serious illnesses are the most worrying [19,22–24].

Meat is a fully interesting subject of study when related to human health, regardless of dietary habits, quantities, or methods of preparation and consumption. A permanent comparison is the one about the way meat is consumed or the type of meat, i.e., species, anatomical region, white or red meat, aquatic originating or terrestrial, etc. Poultry seemed to be healthier than red or processed meat when colorectal cancer was a study subject [25], but consumed as refrigerated, frozen, and as minced meat. In pancreatic carcinoma, an inverse association was shown in poultry meat when compared with red meat [26]. What about when smoked meat is consumed? Traditional and long-lasting smoking of meat preserves its quality with antioxidant and antimicrobial properties, and also imparts a desirable colour, flavour, and aroma to smoked foods [27]. In fish, smoking either cold or hot has a long tradition [28], and PAH concentrations greatly vary among the producers, sometimes exceeding the regulation limits [29]; however, few associations with specific pathologies or genotoxic effects are documented [30]. In poultry, smoking methods vary, e.g., at least four were described at industrial scale [31], while traditional methods imply natural smoke generated from hardwood sawdust. Solutions are being sought for the optimization of technological ways to reduce the risk of PAHs depositing or synthesis in food [32], while smoke flavour is highly desirable in developing societies [33]. Nonetheless, a potential concern for consumers' health and for specific population groups can be easily seen in populations with traditions for consuming smoked food [34].

As a request to health concerns about PAH, European Commission experts are trying to standardize detection methods in tandem with new scientific discoveries [35], while the producers offer a wider range of varieties of smoked meat. Poultry meat is either traditionally or warm-flavour smoked, although liquid smoke could be a viable alternative to traditional methods sensitively appreciated [36], but even as an all-natural antimicrobial in preserving food [27]. In spite of more and more explicit methodology in PAHs detection, not much information is offered about smoking poultry meat, and even less is available for duck meat or turkey. Within this context, we aimed to assess PAH levels in traditional smoking hardwood sawdust applied (birch, plum, cherry wood) to poultry meat (chicken, duck, and turkey).

## 2. Materials and Methods

### 2.1. Biological Material

Chicken (*Gallus gallus domesticus*), turkey (*Meleagris gallopavo*), and ducks (*Anas platyrhynchos*) were traditionally bred and provided the meat used for the experiment. Farmers produce their own reproductive material each year; consequently, no pure-blood, certified breeds were raised. The fowl were reproduced in their own farm, specifically in Posmuș place (a village in Șieu rural community, Bistrița-Năsăud County, Transylvania, Romania, geo coordinates N lat. 46.9753, E long. 24.5814), and were slaughtered in Caraiman Slaughterhouse, Bistrița-Năsăud County, for commercial purposes, to be marketed as traditionally farmed poultry products. A semi-intensive farming system was applied, using locally available feedstuffs during warm season to formulate a maize-soybean diet, following the nutritional requirements of each species. Fowl (30 males from each species) were slaughtered at different ages and live weights: 16 weeks and 3.2 kg/head in chickens, 16 weeks and 3.0 kg/head in ducks, 21 weeks and 6.1 kg in turkey. Breast anatomic region has been used for smoking (*Pectoralis superficialis* and *Pectoralis profundis* muscles).

In terms of research bioethics, no animals were used for applying experimental factors on them within a farm-conducted protocol. They were raised for meat production and sold as eviscerated carcasses to farmers' market. Samples were taken from refrigerated carcasses. No ethical approval was necessary, as the biological material consisted of meat issued from marketed poultry, and the research did not interfere with fowl welfare and farming conditions.

### 2.2. Smoking Materials

Three types of wood were used for smoking: beech (*Fagus sylvatica*), plum (*Prunus domestica*), and cherry (*Prunus cerasus*). All types of wood had the same size of wood chunks (approximate dimensions: $6 \times 2 \times 0.3$ cm) while a continuous maintenance of the hot embers was carried out. Two time intervals were set, one-day smoke treatment (24 h) and two-day smoking treatment (48 h). A traditional wooden, handmade installation was used, and a natural velocity of smoke through a metal sieve covered pipe of $20 \times 20$ cm occurred.

### 2.3. Meat Proximate Composition

Breast pieces of 10 g were taken from all individuals, prior to smoking, then were minced and blended together to form a homogenous sample of 300 g/species, thereafter submitted to analytical chemistry investigation to assess the proximate composition. Most of the methods involved an initial gravimetric assessment of a crude sample using a KERN ABJ 220 4NM analytical scale (manufacturer KERN & SOHN GmbH, Stuttgart, Germany). AOAC standard protocols were used to assess proximate composition: dry matter (DM)—water content via AOAC 950.46 method [37], on a thermoregulated forced airflow MEMMERT UF110+ oven (manufacturer Memmert GmbH + Co., Schwabach, Germany); total minerals (TM) content via AOAC 920.153 method [38], on a Nabertherm muffle furnace L 9/13 (manufacturer Nabertherm GmbH, Lilienthal, Germany); total lipids (TL) content—ether extract via AOAC 960.39 method [39], on a Velp Scientific SER 148 Randall extractor (manufacturer VELP Scientifica SRL, Usmate, Italy); total nitrogen

content and total protein (TP) content via AOAC 929.08 method [40], on a Velp Scientifica DK6 digester and a UDK127-distiller (manufacturer VELP Scientifica SRL, Usmate, Italy) and Biotrate 50 mL digital burette (manufacturer Sartorius Lab Instruments GmbH & Co., Goettingen, Germany). Total organic matter (TOM) was calculated by the difference between Dry Matter (DM) and Total minerals (TM) content (Relation (1)).

$$\text{TOM (g/100 g)} = \text{DM (g/100 g)} - \text{TM (g/100 g)} \tag{1}$$

Nitrogen-Free Extract (NFE) was calculated by the difference between Organic Matter content and Total lipids and Total protein content (Relation (2)).

$$\text{NFE (g/100 g)} = \text{DM (g/100 g)} - \text{TL (g/100 g)} - \text{TP (g/100 g)} \tag{2}$$

For each proximate compound and each derived computation, 6 analytical repetitions were run. All results were expressed as grams per 100 g crude sample.

### 2.4. HPLC Analysis

Extraction of polycyclic aromatic hydrocarbons (PAHs) from smoked, skinless chicken, turkey, and duck breast meat started from an average sample of 10 g/species, homogenized in a blender, and saponified in an ultrasound bath for 30 min at 60 °C, using 50 mL KOH 0.4 M solution in ethanol and water in a 9:1 ratio. Post saponification, the samples were passed through sodium sulphate and filtered, then a double extraction was performed using 15 mL cyclohexane. The supernatant was reunited and purified on a Florisil column, then evaporated to dryness under a continuous nitrogen atmosphere. Each sample was dissolved in acetonitrile (1 mL) and filtered through a 0.45 μm filter, then kept in the refrigerator until the beginning of chromatographic analysis. This was carried out on a Perkin-Elmer HPLC system (manufacturer Perkin-Elmer, equipped with a model 200 series binary pump delivery system (Perkin-Elmer), a model 200 series degasser (manufacturer Perkin-Elmer Inc., Boston, MA, USA), a Flexar autosampler (Perkin-Elmer), a Spark thermostat, and a fluorescence detector model 200 series (Perkin-Elmer) with a double monochromator. The identification of PAHs was carried out on a column of ZORBAX Eclipse PAH (5 μm, 15 cm × 4.6 mm) in a gradient consisting of acetonitrile and ultrapure water as the mobile phase.

The LOQ and LOD were calculated as 10 times and 3.3 times, respectively, the standard deviation of the response for the lowest feasible analytic concentration in the range and the calibration curve's slope. Calibration curve 0.05–10 ng/g. Limit of quantification 0.05 ng/g. Limit of detection 0.015 ng/g. Recovery between 88.3–97.2 %, calculated using spiked samples with a concentration of 5 ng/g.

Fifteen PAHs were aimed for detection: Benzo(a)pyrene, Naphthalene, Acenaphthene, Flourene, Phenanthrene, Anthracene, Fluoranthene, Pyrene, Benz(a)anthracene, Chrysene, Benzo(k)fluoranthene, Benzo(b)fluoranthene, Benzo[*ghi*]perylene, Dibenzo[a,h]anthracene, and Indeno [1,2,3-*cd*]pyrene. Eleven PAHs were utterly quantified among the 15. All acquired data resulted after 5 analytical replications for each PAH.

The sum of benzo(a)pyrene, benz(a)anthracene, benzo(b)fluoranthene, and chrysene (level of Σ4PAHs in smoked meat) was also quantified, representing an index for evaluating the toxicity induced by PAHs on meat quality, according to EU legislation.

### 2.5. Statistical Analyses

Analytical data were statistically processed via Graph Pad Prism 9.4.1 (manufacturer GraphPad Inc., Palo Alto, CA, USA) software to achieve the descriptive statistics values (mean, standard error of mean, coefficient of variation) and to analyse the variance using the ANOVA one-way algorithm, followed by Tukey post-hoc treatment, when 3 data groups were compared, or the unpaired 2-tailed *t* test when just 2 data groups were compared. The results of the analysis of variance were reported as *p* values, compared to *p* thresholds of 0.05, 0.01, and 0.001.

Within the same fowl species, meat samples were compared for differences in PAHs accumulation at 24 and 48 h post-smoking, in relation to smoking materials (burnt wood). Additionally, comparisons were carried out between the fowl species, within the same smoking type, to find out whether meat structure and its other particularities affect the accumulation of the same PAHs.

## 3. Results

The proximate composition of meat, prior to smoking, is displayed in Table 1. Fat content in meat, as the main accumulation matrix for the investigated PAH, varied from 1.88 g/100 g in chicken breast to 2.42 g/100 g in turkey and to 2.81 g/100 g in duck breast meat (Table 1).

**Table 1.** Proximate composition of the matured poultry meat prior to smoking (g/100 g).

| Proximate Compound | Chicken | | Turkey | | Duck | |
|---|---|---|---|---|---|---|
| | Mean ± SEM | V% | Mean ± SEM | V% | Mean ± SEM | V% |
| Water | 77.35 ± 0.55 | 1.75 | 73.28 ± 0.64 | 2.13 | 81.27 ± 0.67 | 2.01 |
| Dry matter | 22.65 ± 0.55 | 5.91 | 26.72 ± 0.64 | 5.86 | 18.73 ± 0.67 | 8.79 |
| Total minerals | 0.96 ± 0.02 | 5.12 | 1.31 ± 0.02 | 4.28 | 0.87 ± 0.02 | 4.81 |
| Organic matters | 21.69 ± 0.39 | 4.38 | 25.41 ± 0.48 | 4.66 | 17.86 ± 0.39 | 5.32 |
| Total proteins | 19.56 ± 0.27 | 3.44 | 22.62 ± 0.35 | 3.83 | 14.74 ± 0.25 | 4.09 |
| Total lipids | 1.88 ± 0.04 | 4.97 | 2.42 ± 0.05 | 5.08 | 2.81 ± 0.06 | 5.19 |
| Nitrogen free extract | 0.25 ± 0.01 | 5.63 | 0.37 ± 0.01 | 5.45 | 0.31 ± 0.01 | 5.81 |

These values of meat nutrients are generally expected when a traditional manner of farming is rolled on, in respect to feeding based mostly on locally available feedstuffs, such as corn and wheat crumbles (concentrated in carbohydrates and, subsequently, energy), and a relatively long period of fattening, compared to conventionally reared broilers, whose diets are based on a corn-soymeal mixture (better balanced diets in terms of energy/protein ratios).

Data on the accumulation of PAHs in chicken meat are presented in Table 2 with comparisons between the type of woods used in smoking and between the duration of exposure.

As expected, the two-day smoking period produced a higher accumulation of PAHs in meat; some of the PAHs even more than doubled in concentration (i.e., Benz(a)anthracene in beech wood smoking or Benzo(k)fluoranthene in cherry wood smoking). Benzo(k)fluoranthene has only been quantified in cherry wood smoking.

Data on the accumulation of PAHs in turkey meat are displayed in Table 3 with differences analysis related to the species of tree sawdust used in smoking and to the smoking length.

**Table 2.** Variance of total PAHs accumulation in chicken meat (µg PAH/kg meat, ppm).

| Polycyclic Aromatic Hydrocarbons | Exposure Time | Beech Wood Smoking (B) | | Plum Wood Smoking (P) | | Cherry Wood Smoking (C) | | Regulated Maximal Admitted Limit |
|---|---|---|---|---|---|---|---|---|
| | | Mean ± SEM | V% | Mean ± SEM | V% | Mean ± SEM | V% | |
| Naphtalene | 24 h | $6.47\ ^a_x \pm 0.14$ | 4.4 | $4.01\ ^{da}_x \pm 0.09$ | 4.5 | $6.74\ ^{ad}_x \pm 0.24$ | 7.1 | At lowest as possible [10] |
| | 48 h | $8.24\ ^a_z \pm 0.21$ | 5.8 | $4.66\ ^{da}_x \pm 0.05$ | 2.2 | $9.07\ ^{ad}_w \pm 0.43$ | 9.4 | |
| Acenaphthene | 24 h | $8.95\ ^a \pm 0.15$ | 3.3 | $6.46\ ^{da} \pm 0.30$ | 9.4 | $10.24\ ^{ad} \pm 0.25$ | 4.9 | At lowest as possible [10] |
| | 48 h | $10.24\ ^a \pm 0.21$ | 4.2 | $7.88\ ^{da} \pm 0.57$ | 14.5 | $11.32\ ^{ad} \pm 0.22$ | 3.8 | |
| Fluorene | 24 h | $32.93\ ^a_x \pm 0.65$ | 3.9 | $16.24\ ^{da}_x \pm 0.40$ | 4.9 | $20.7\ ^{dd}_x \pm 0.32$ | 3.1 | At lowest as possible [10] |
| | 48 h | $37.02\ ^a_w \pm 0.31$ | 1.7 | $18.56\ ^{da}_z \pm 0.17$ | 1.9 | $22.53\ ^{dd}_y \pm 0.23$ | 2.1 | |
| Phenanthrene | 24 h | $54.85\ ^a_x \pm 0.59$ | 2.2 | $42.45\ ^d_x \pm 1.09$ | 5.1 | $43.68\ ^d_x \pm 0.52$ | 2.4 | At lowest as possible [10] |
| | 48 h | $59.60\ ^a_y \pm 0.99$ | 3.3 | $64.53\ ^{ca}_w \pm 0.39$ | 1.2 | $77.16\ ^{dd}_w \pm 1.07$ | 2.8 | |
| Anthracene | 24 h | $2.01\ ^a_x \pm 0.03$ | 3.3 | $0.84\ ^d \pm 0.04$ | 8.8 | $0.70\ ^d_x \pm 0.03$ | 9.3 | At lowest as possible [10] |
| | 48 h | $3.22\ ^a_w \pm 0.11$ | 6.7 | $0.96\ ^d \pm 0.03$ | 6.3 | $0.94\ ^d_y \pm 0.02$ | 3.8 | |
| Fluoranthene | 24 h | $6.05\ ^a_x \pm 0.08$ | 2.6 | $4.09\ ^{da}_x \pm 0.20$ | 9.9 | $2.51\ ^{dd}_x \pm 0.20$ | 16.1 | At lowest as possible [10] |
| | 48 h | $7.18\ ^a_w \pm 0.11$ | 2.9 | $5.30\ ^{da}_w \pm 0.17$ | 6.4 | $4.09\ ^{dd}_w \pm 0.12$ | 5.9 | |
| Pyrene | 24 h | $0.88\ ^a \pm 0.03$ | 6.4 | $1.48\ ^{ca} \pm 0.07$ | 8.9 | $2.18\ ^{dd}_x \pm 0.13$ | 11.5 | At lowest as possible [10] |
| | 48 h | $1.17\ ^a \pm 0.11$ | 18.3 | $1.82\ ^{da} \pm 0.04$ | 4.8 | $3.52\ ^{dd}_w \pm 0.12$ | 6.8 | |
| Benz(a)anthracene | 24 h | $0.32\ ^a_x \pm 0.02$ | 15.1 | $0.68\ ^{aa} \pm 0.07$ | 19.9 | $1.65\ ^{dd} \pm 0.13$ | 16.2 | At lowest as possible [10] |
| | 48 h | $0.77\ ^a_y \pm 0.05$ | 12.1 | $0.95\ ^{aa} \pm 0.05$ | 11.4 | $1.76\ ^{dd} \pm 0.15$ | 16.9 | |
| Chrysene | 24 h | $0.98\ ^a_x \pm 0.03$ | 5.5 | $1.70\ ^{da} \pm 0.05$ | 6.3 | $0.70\ ^{ad}_x \pm 0.06$ | 17.4 | Σ4PAHs < 30 µg/kg [10] |
| | 24 h | $1.59\ ^a_z \pm 0.125$ | 15.3 | $2.08\ ^b \pm 0.12$ | 11.3 | $1.69\ ^{ab}_w \pm 0.16$ | 20.3 | |
| Benzo(k)fluoranthene | 24 h | - | - | - | - | $0.75\ _x \pm 0.07$ | 18.4 | At lowest as possible [10] |
| | 48 h | - | - | - | - | $1.88\ _w \pm 0.03$ | 2.9 | |

SEM = Standard error of mean; V% = coefficient of variation. Analysis of variance: per line, means with different superscripts on the same line differ significantly for: $^{ab}$ $p < 0.05$; $^{ac}$ $p < 0.01$; $^{ad}$ $p < 0.001$; per column (24 h vs. 48 h), means with different subscripts on the same column differ significantly for: $_{xy}$ $p < 0.05$; $_{xz}$ $p < 0.01$; $_{xw}$ $p < 0.001$. Σ4PAHs: Sum of benzo(a)pyrene, benz(a)anthracene, benzo(b)fluoranthene, and chrysene is <30 µg/kg in smoked meat and meat products [10].

**Table 3.** Variance of total PAHs accumulation in turkey meat (µg PAH/kg meat, ppm).

| Polycyclic Aromatic Hydrocarbons | Exposure Time | Beech Wood Smoking (B) | | Plum Wood Smoking (P) | | Cherry Wood Smoking (C) | | Regulated Maximal Admitted Limit |
|---|---|---|---|---|---|---|---|---|
| | | Mean ± SEM | V% | Mean ± SEM | V% | Mean ± SEM | V% | |
| Naphtalene | 24 h | $4.85\ ^a \pm 0.14$ | 5.82 | $3.37\ ^c_x \pm 0.26$ | 15.19 | $4.14\ _x \pm 0.33$ | 15.86 | At lowest as possible [10] |
| | 48 h | $5.37 \pm 0.22$ | 8.04 | $5.06\ _w \pm 0.24$ | 9.41 | $5.33\ _y \pm 0.12$ | 4.68 | |
| Acenaphthene | 24 h | $7.93\ ^a_x \pm 0.29$ | 7.36 | $6.19\ ^{ba} \pm 0.36$ | 11.48 | $9.97\ ^{cd}_x \pm 0.53$ | 10.56 | At lowest as possible [10] |
| | 48 h | $9.60\ ^a_y \pm 0.30$ | 6.27 | $7.74\ ^{ba} \pm 0.33$ | 8.42 | $11.79\ ^{cd}_y \pm 0.39$ | 6.59 | |
| Fluorene | 24 h | $23.96\ ^a_x \pm 0.89$ | 7.43 | $17.51\ ^d_x \pm 0.39$ | 4.48 | $18.45\ ^d_x \pm 0.37$ | 3.99 | At lowest as possible [10] |
| | 48 h | $29.91\ ^a_w \pm 0.61$ | 4.06 | $21.51\ ^d_z \pm 1.02$ | 9.46 | $23.78\ ^d_w \pm 0.56$ | 4.71 | |
| Phenanthrene | 24 h | $46.45\ ^a \pm 1.23$ | 5.32 | $32.82\ ^d_x \pm 1.16$ | 7.05 | $46.70\ ^a_x \pm 3.00$ | 12.85 | At lowest as possible [10] |
| | 48 h | $53.58\ ^a \pm 1.21$ | 4.52 | $44.50\ ^{ba}_x \pm 1.72$ | 7.72 | $57.70\ ^{ad}_z \pm 2.25$ | 7.79 | |
| Anthracene | 24 h | $1.75\ ^a_x \pm 0.09$ | 9.85 | $0.76\ ^d \pm 0.07$ | 17.66 | $0.51\ ^d \pm 0.03$ | 13.32 | At lowest as possible [10] |
| | 48 h | $2.40\ ^a_y \pm 0.29$ | 23.78 | $0.98\ ^d \pm 0.08$ | 16.74 | $0.72\ ^d \pm 0.08$ | 23.68 | |
| Fluoranthene | 24 h | $2.64\ ^a \pm 0.23$ | 17.39 | $4.27\ ^{ca}_x \pm 0.34$ | 15.89 | $1.68\ ^{cd} \pm 0.18$ | 21.36 | At lowest as possible [10] |
| | 48 h | $3.82\ ^a \pm 0.49$ | 25.92 | $5.57\ ^{ca}_y \pm 0.20$ | 7.31 | $2.31\ ^{bd} \pm 0.13$ | 10.88 | |
| Pyrene | 24 h | $0.61\ ^a_x \pm 0.06$ | 18.45 | $0.96\ ^a \pm 0.05$ | 10.40 | $2.05\ ^d \pm 0.09$ | 8.89 | At lowest as possible [10] |
| | 48 h | $1.23\ ^a_z \pm 0.18$ | 29.76 | $1.07\ ^a \pm 0.16$ | 6.45 | $2.45\ ^d \pm 0.08$ | 11.91 | |
| Benz(a)anthracene | 24 h | $0.63 \pm 0.04$ | 11.91 | $0.69 \pm 0.08$ | 17.21 | $0.60\ _x \pm 0.05$ | 12.70 | At lowest as possible [10] |
| | 48 h | $0.83 \pm 0.05$ | 12.70 | $0.80 \pm 0.10$ | 16.33 | $0.92\ _y \pm 0.08$ | 17.29 | |
| Chrysene | 24 h | $0.71\ ^a \pm 0.06$ | 17.29 | $1.01\ ^{ba} \pm 0.02$ | 8.50 | $1.77\ ^{dd}_x \pm 0.08$ | 10.40 | Σ4PAHs < 30 µg/kg [10] |
| | 48 h | $0.89\ ^a \pm 0.05$ | 10.40 | $1.06\ ^{da} \pm 0.05$ | 8.69 | $2.19\ ^{dd}_z \pm 0.10$ | 15.74 | |
| Benzo(k)fluoranthene | 24 h | - | - | - | - | $0.38\ _x \pm 0.16$ | 32.76 | At lowest as possible [10] |
| | 48 h | - | - | - | - | $0.71\ _z \pm 0.17$ | 19.57 | |
| Benzo(a)pyrene | 24 h | - | - | - | - | $0.51 \pm 0.04$ | 15.74 | < 5 µg/kg [10] |
| | 48 h | - | - | - | - | $0.59 \pm 0.05$ | 17.16 | |

SEM = Standard error of mean; V% = coefficient of variation. Analysis of variance: per line, means with different superscripts on the same line differ significantly for: $^{ab}$ $p < 0.05$; $^{ac}$ $p < 0.01$; $^{ad}$ $p < 0.001$; per column (24 h vs. 48 h), means with different subscripts on the same column differ significantly for: $_{xy}$ $p < 0.05$; $_{xz}$ $p < 0.01$; $_{xw}$ $p < 0.001$. Σ4PAHs: Sum of benzo(a)pyrene, benz(a)anthracene, benzo(b)fluoranthene, and chrysene is <30 µg/kg in smoked meat and meat products [10].

In turkey meat, the two-day smoking also quantified more PAHs, with Pyrene doubling the concentration in beech wood smoking from 0.61 µg after day 1 to 1.23 µg after two smoking days. Overall, the absolute values of the PAHs were lower in turkey meat as compared to chicken meat. The sum of Benz(a)anthracene and Chrysene (two of the set of four PAHs that were monitoring markers of PAHs toxicity accumulation in meat based on EU legislation) was similar after day 1 in chicken and turkey meat (2.35 µg vs. 2.37 µg, respectively), but the differences were slightly in favour of chicken meat after day 2 of smoking (3.45 µg vs 3.11 µg, respectively).

The dynamics of PAHs deposition and accumulation in duck meat in relation to the duration of smoking and the species of tree used in smoking are presented in Table 4.

**Table 4.** Variance of total PAHs accumulation in duck meat (µg PAH/kg meat, ppm).

| Polycyclic Aromatic Hydrocarbons | Exposure Time | Beech Wood Smoking (B) | | Plum Wood Smoking (P) | | Cherry Wood Smoking (C) | | Regulated Maximal Admitted Limit |
|---|---|---|---|---|---|---|---|---|
| | | Mean ± SEM | V% | Mean ± SEM | V% | Mean ± SEM | V% | |
| Naphtalene | 24 h | 8.55 [a] ± 0.25 | 5.93 | 6.46 [da]$_x$ ± 0.23 | 7.27 | 9.42 [ad]$_x$ ± 0.26 | 5.50 | At lowest as possible [10] |
| | 48 h | 9.53 [a] ± 0.46 | 9.66 | 8.76 [aa]$_w$ ± 0.13 | 2.89 | 11.11 [cd]$_z$ ± 0.41 | 7.42 | |
| Acenaphthene | 24 h | 9.10 $_x$ ± 0.44 | 9.71 | 8.94 ± 0.13 | 3.17 | 10.24 $_x$ ± 0.39 | 7.68 | At lowest as possible [10] |
| | 48 h | 10.81 [a]$_y$ ± 0.21 | 3.88 | 10.40 [aa] ± 0.30 | 5.70 | 12.26 [bc]$_z$ ± 0.42 | 6.93 | |
| Fluorene | 24 h | 27.12 ± 1.23 | 9.09 | 25.80 ± 0.52 | 4.02 | 24.60 ± 0.83 | 6.73 | At lowest as possible [10] |
| | 48 h | 31.20 ± 1.36 | 8.72 | 29.26 ± 0.71 | 4.88 | 27.80 ± 0.86 | 6.19 | |
| Phenanthrene | 24 h | 84.74 [a] ± 1.95 | 4.59 | 74.16 [ba]$_x$ ± 3.04 | 8.21 | 71.56 [ca]$_x$ ± 3.11 | 8.70 | At lowest as possible [10] |
| | 48 h | 93.24 ± 1.20 | 2.57 | 94.00 $_w$ ± 1.41 | 3.01 | 88.06 $_w$ ± 2.12 | 4.82 | |
| Anthracene | 24 h | 3.19 [a]$_x$ ± 0.21 | 12.99 | 0.92 [da]$_x$ ± 0.10 | 20.36 | 2.19 [dd]$_x$ ± 0.22 | 20.44 | At lowest as possible [10] |
| | 48 h | 4.06 [a]$_z$ ± 0.14 | 6.79 | 1.62 [da]$_y$ ± 0.09 | 11.39 | 2.84 [dd]$_y$ ± 0.04 | 2.86 | |
| Fluoranthene | 24 h | 8.84 [a] ± 0.45 | 5.09 | 6.38 [d]$_x$ ± 0.67 | 10.51 | 5.40 [d] ± 0.67 | 12.45 | At lowest as possible [10] |
| | 48 h | 9.73 [a] ± 0.22 | 4.08 | 7.67 [da]$_y$ ± 0.34 | 8.81 | 6.22 [db] ± 0.34 | 5.25 | |
| Pyrene | 24 h | 1.48 [a]$_x$ ± 0.14 | 18.89 | 2.10 [aa] ± 0.14 | 13.28 | 2.56 [db]$_x$ ± 0.20 | 15.74 | At lowest as possible [10] |
| | 48 h | 2.17 [a]$_y$ ± 0.14 | 12.68 | 2.50 [aa] ± 0.18 | 14.81 | 3.26 [db]$_y$ ± 0.11 | 6.87 | |
| Benz(a)anthracene | 24 h | 0.94 [a] ± 0.08 | 17.21 | 1.08 [aa]$_x$ ± 0.15 | 28.33 | 1.68 [cb] ± 0.09 | 10.24 | At lowest as possible [10] |
| | 48 h | 1.38 [a] ± 0.16 | 22.88 | 1.74 [aa]$_z$ ± 0.11 | 12.90 | 2.12 [ca] ± 0.07 | 6.37 | |
| Chrysene | 24 h | 1.88 [a] ± 0.31 | 22.05 | 2.63 [ba] ± 0.10 | 7.37 | 1.78 [ac]$_x$ ± 0.10 | 11.40 | Σ4PAHs < 30 µg/kg [10] |
| | 48 h | 2.12 [a] ± 0.16 | 14.74 | 3.10 [da] ± 0.06 | 3.98 | 2.48 [ab]$_y$ ±0.17 | 13.59 | |
| Benzo(k)fluoranthene | 24 h | - | - | - | - | 0.89 $_x$ ± 0.07 | 16.54 | At lowest as possible [10] |
| | 48 h | - | - | - | - | 2.07 $_w$ ± 0.09 | 8.62 | |

SEM = Standard error of mean; V% = coefficient of variation. Analysis of variance: per line, means with different superscripts on the same line differ significantly for: [ab] $p < 0.05$; [ac] $p < 0.01$; [ad] $p < 0.001$; per column (24 h vs. 48 h), means with different subscripts on the same column differ significantly for: $_{xy}$ $p < 0.05$; $_{xz}$ $p < 0.01$; $_{xw}$ $p < 0.001$. Σ4PAHs: Sum of benzo(a)pyrene, benz(a)anthracene, benzo(b)fluoranthene, and chrysene is <30 µg/kg in smoked meat and meat products [10].

As shown in Table 3, the PAHs accumulation in duck meat increased from day 1 to day 2 in all smoking wood types, the highest being in cherry wood smoking for Benzo(k)fluoranthene, with more than doubling the concentration (from 0.89 µg to 2.07 µg). The most relevant finding is that the absolute values summarized for all PAHs are highest in duck meat when compared with chicken and turkey meat. In addition, the sum of Benz(a)anthracene and Chrysene was higher for duck meat, irrespective of the day of smoking, indicating similar values for day 1 with day 2 for chicken and turkey meat (3.46 µg in duck meat on day 1 vs. 3.45 µg for chicken meat and 3.11 µg for turkey meat, but the last two after two days of smoking). After two days of smoking, duck meat accumulated 4.6 µg of Benz(a)anthracene + Chrysene.

Table 5 reveals the influence of meat origin on the dynamics of PAHs accumulation, in relation to exposure time to smoking, within each type of fuel used to generate smoke.

**Table 5.** Analysis of variance of total PAHs accumulation in meat, under the influence of poultry species (arithmetic means, µg PAH/kg meat, ppm).

| Polycyclic Aromatic Hydrocarbons | Exposure Time | Beech Wood Smoking (B) | | | Plum Wood Smoking (P) | | | Cherry Wood Smoking (C) | | | Regulated Maximal Admitted Limit |
|---|---|---|---|---|---|---|---|---|---|---|---|
| | | Chicken | Turkey | Duck | Chicken | Turkey | Duck | Chicken | Turkey | Duck | |
| Naphtalene | 24 h | 6.47 [a] | 4.85 [ca] | 8.55 [dd] | 4.01 [a] | 3.37 [a] | 6.46 [d] | 6.74 [a] | 4.14 [da] | 9.42 [d] | At lowest as possible [10] |
| | 48 h | 8.24 [a] | 5.37 [da] | 9.53 [ad] | 4.66 [a] | 5.06 [a] | 8.76 [d] | 9.07 [a] | 5.33 [da] | 11.11 [d] | |
| Acenaphthene | 24 h | 8.95 | 7.93 | 9.10 | 6.46 [a] | 6.19 [a] | 8.94 [d] | 10.24 | 9.97 | 10.24 | At lowest as possible [10] |
| | 48 h | 10.24 | 9.60 | 10.81 | 7.88 [a] | 7.74 [a] | 10.40 [d] | 11.32 | 11.79 | 12.26 | |
| Fluorene | 24 h | 32.93 [a] | 23.96 [d] | 27.12 [d] | 16.24 [a] | 17.51 [a] | 25.80 [d] | 20.70 [a] | 18.45 [aa] | 24.60 [bd] | At lowest as possible [10] |
| | 48 h | 37.02 [a] | 29.91 [d] | 31.20 [d] | 18.56 [a] | 21.51 [a] | 29.26 [d] | 22.53 [a] | 23.78 [aa] | 27.80 [db] | |
| Phenanthrene | 24 h | 54.85 [a] | 46.45 [a] | 84.74 [d] | 42.45 [a] | 32.82 [ba] | 74.16 [d] | 43.68 [a] | 46.70 [a] | 71.56 [d] | At lowest as possible [10] |
| | 48 h | 59.60 [a] | 53.58 [a] | 93.24 [d] | 64.53 [a] | 44.50 [a] | 94.00 [d] | 77.16 [a] | 57.70 [da] | 88.06 [cd] | |
| Anthracene | 24 h | 2.01 [a] | 1.75 [a] | 3.19 [d] | 0.84 | 0.76 | 0.92 | 0.70 [a] | 0.51 [a] | 2.19 [d] | At lowest as possible [10] |
| | 48 h | 3.22 [a] | 2.40 [da] | 4.06 [dd] | 0.96 [a] | 0.98 [a] | 1.62 [b] | 0.94 [a] | 0.72 [a] | 2.84 [d] | |
| Fluoranthene | 24 h | 6.05 [a] | 2.64 [da] | 8.84 [dd] | 4.09 [a] | 4.27 [aa] | 6.38 [da] | 2.51 [a] | 1.68 [a] | 5.40 [d] | At lowest as possible [10] |
| | 48 h | 7.18 [a] | 3.82 [da] | 9.73 [dd] | 5.30 [a] | 5.57 [a] | 7.67 [d] | 4.09 [a] | 2.31 [da] | 6.22 [d] | |
| Pyrene | 24 h | 0.88 [a] | 0.61 [a] | 1.48 [d] | 1.48 [a] | 0.96 [aa] | 2.10 [ad] | 2.18 | 2.05 | 2.56 | At lowest as possible [10] |
| | 48 h | 1.17 [a] | 1.23 [a] | 2.17 [d] | 1.82 [a] | 1.07 [ca] | 2.50 [bd] | 3.52 [a] | 2.45 [d] | 3.26 [a] | |
| Benz(a)anthracene | 24 h | 0.32 [a] | 0.63 [aa] | 0.94 [ca] | 0.68 | 0.69 | 1.08 | 1.65 [a] | 0.60 [d] | 1.68 [a] | At lowest as possible [10] |
| | 48 h | 0.77 [a] | 0.83 [aa] | 1.38 [cc] | 0.95 [a] | 0.80 [a] | 1.74 [d] | 1.76 [a] | 0.92 [d] | 2.12 [a] | |
| Chrysene | 24 h | 0.98 [a] | 0.71 [a] | 1.88 [d] | 1.70 [a] | 1.01 [ca] | 2.63 [dd] | 0.70 [a] | 1.77 [d] | 1.78 [d] | Σ4PAHs < 30 µg/kg [10] |
| | 48 h | 1.59 [a] | 0.89 [ca] | 2.12 [ad] | 2.08 [a] | 1.06 [da] | 3.10 [dd] | 1.69 [a] | 2.19 [aa] | 2.48 [da] | |
| Benzo(k)fluoranthene | 24 h | - | - | - | - | - | - | 0.75 [a] | 0.38 [ca] | 0.89 [ad] | At lowest as possible [10] |
| | 48 h | - | - | - | - | - | - | 1.88 [a] | 0.71 [d] | 2.07 [a] | |
| Benzo(a)pyrene | 24 h | - | - | - | - | - | - | - | 0.51 | - | < 5 µg/kg [10] |
| | 48 h | - | - | - | - | - | - | - | 0.59 | - | |

Analysis of variance: per line, means with different superscripts on the same line differ significantly for: [ab] $p < 0.05$; [ac] $p < 0.01$; [ad] $p < 0.001$; Σ4PAHs: Sum of benzo(a)pyrene, benz(a)anthracene, benzo(b)fluoranthene, and chrysene is <30 µg/kg in smoked meat and meat products [10].

Table 5 shows comparatively how the PAHs varied between species, based on smoking wood and period of smoking. Overall, the duck meat has the higher PAHs accumulation, although there is an exception for Fluorene when beech wood was used. Chicken meat presented higher values for day 1 and day 2, followed by duck and turkey meat, respectively. Significant differences ($p < 0.001$) were usually observed for duck meat when compared with chicken and turkey meat, but also easy to notice as absolute values for Anthracene, Phenanthrene, or Fluoranthene.

## 4. Discussion

The concentration limits were found below the EU legislation [41] thresholds for benzo(a) pyrene (BaP), for the sum of benzo(a)pyrene, benz(a)anthracene, benzo(b)fluoranthene, and chrysene (level of Σ4PAHs in smoked meat should not exceed 12 µg/kg or 30 µg/kg for some of the member state countries since 13.12.2014 [10]) in all three poultry species. Those four PAHs are highly indicative and carefully monitored when present in food. The highest concentrations of PAHs occurred in duck meat, irrespective of time of smoking or wood type (Table 4), followed by chicken meat (Table 2) and turkey (Table 3). Duck meat also accumulated PAHs faster, as the percent of total PAHs at the end of the first day of smoking per total period.

In comparison, Peking roasted ducks in the hung oven roasted technique had a high BaP concentration of 8.7 µg/kg in the skin, but <3.0 µg/kg when other techniques, such as the closed oven procedure and electricity heating, were used [42]. Within the same study, ducks roasted by electrical heating presented lower concentrations of total PAHs. No BaP were detected in smoked duck meat, but higher Σ4PAHs were, followed by chicken and turkey meat. Duck meat reached the highest value for chrysene (one of the Σ4PAHs with legislative imposition limits) of 3.10 µg/kg when plum wood and two days of smoking treatment was used. T1hat represents 10.33% of the Σ4PAHs. When smoking time was

used as reference, the duck meat accumulated the highest concentration of chrysene after only one day of smoking, with a value of 2.63 μg/kg (again in plum wood smoking), which was superior to all other meats after two days of smoking. This might correlate with the fat percentages of carcasses. Our samples had no skin at all. Generally, fatter meat accumulates higher PAH concentrations [43,44], while proteins and carbohydrates do not contribute to the formation of PAHs during thermal cooking [43]. Fat distribution inside the breast muscle differs among species, but usually a low percentage is found in chicken and higher in duck [45]. This fat is generally uniformly distributed within the muscle and surrounding muscle fibres.

The skin might significantly contribute to PAHs accumulation, but as a natural "barrier" too. In pork meat, skin has been suggested to protect the bacon from PAHs concentration [46]. Added fat in Portuguese traditional dry fermented sausages positively influenced PAHs contamination [47]. PAH compounds were found to be more abundant in high-fat doner samples than in low-fat groups when fat content was interrogated [48], while a positive linear correlation was observed for the PAHs' bio-accessibility and the fat contents of grilled meat [49]. That is also observed in various traditionally smoked products from Cyprus, where the highest PAH concentrations were found in samples with higher fat content [50]. The fat content of the meat products appears to favour PAHs accumulation in smoking products. Normally, smoked meat is left with the skin on, knowing that the fat gives it superior organoleptic qualities. There is also a small risk of dehydration of smoked meat when the skin is removed, but a short smoking period could help to avoid this phenomenon.

The tissue composition and tissue texture of the meat are other characteristics that could influence the smoking process. Tissue texture in poultry species varies depending on several factors [51], but genetic ones mostly differentiate the chemical and tissue composition of the meat from the three species. Meat texture and intramuscular connective tissue play an important role in meat quality [52], and texture affects bio-concentrating and PAHs accumulation properties [13]. In our sample data, turkey meat has the lowest total content of PAHs and also Σ4PAHs. Interestingly, benzo(a)pyrene was only quantified in low quantities (average of 0.512 μg/kg in the single-day smoking treatment and average of 0.592 μg/kg in the two-day smoking treatment) in turkey meat and only in cherry wood smoking. From this perspective, the values are far lower than the legislative imposition limit of 5 μg/kg. Anyway, the presence of a highly monitored PAH only in turkey meat is somehow surprising based on the chemical composition of the three types of meat used in the experiment. Duck meat has accumulated the highest concentrations overall of PAHs, but no benzo(a)pyrene. Possibly, the fat content was one of the most significant factors in PAHs accumulation, but there are certainly others to be interrogated. Duck meat has the highest fat content of the meat, while turkey exhibited the highest protein content (22.62% from dry matter vs 14.74% in duck meat in our experiment). Chicken meat had intermediate values with respect to both fat and protein content.

The type of wood used for smoke generated significant differences ($p < 0.01$) for the same smoking period (one day or two days). Irrespective of meat species, the highest concentration overall (average of the three poultry species) of PAHs was quantified when beech wood was used for smoking, followed by cherry and plum woods (Tables 1–3). Taking the poultry species separately, the same order was recorded (when averaging day one and day two). Beech wood is largely used in smoking meat and is part of Romanian tradition. It also generated higher concentrations of PAHs in various pork meat products, e.g., Frankfurter-type sausages and mini-salamis when compared with several other wood types in an experiment where a smoking chamber with a smoldering smoke generator was used [14,53].

Interestingly, only cherry wood generated benzo(k)fluoranthene in all species when the two-day smoking period occurred, and benzo(a)pyrene in duck meat, although in low concentrations (Tables 1–4). Bird-cherry wood has been shown to generate moderate to low concentrations of PAHs in pork meat smoked in a system of homemade smoking kiln,

when 10 wood types and charcoal were used [54]. The same was not true for chicken and turkey meat, where plum wood utilization produced a significantly lower accumulation of PAHs when compared with both beech and cherry wood ($p < 0.05$).

Among the three species, duck recorded the fastest PAHs accumulation in meat, as the day one percentage from the final smoking time (83.95% as average for all three types of wood vs. 79.91% in turkey and 75.98% in poultry).

Surprisingly, the highest concentration of PAHs as the percent of accumulation from day one through day two was in chicken meat, although it has a low-fat percent in meat as compared with duck and turkey meat.

Benzo(a)pyrene, although present in turkey meat, was only a little concentrated ($p > 0.05$) with the doubling of the smoking time. The first smoking period has been shown to significantly accumulate higher PAH concentrations in grilled beef and pork meat, in direct correlation with fat percent and fat dripping from the meat samples onto the heat source during grilling [44]. In our experiment, fat dripping was excluded due to the smoking technique itself.

From PAHs, the higher concentration was observed for benz(a)anthracene, chrysene, and benzo(k)fluoranthene in chicken meat after 2 days of smoking. More than doubled values were generated by cherry wood for chrysene and benzo(k)fluoranthene and for benz(a)anthracene by smoking with beech wood.

In duck meat, the total average values of PAHs revealed non-significant differences for all types of wood after two days smoking treatment, suggesting a mild or any dependency of this parameter. This also might be interpreted as the high capacity of PAHs accumulation with increasing time. A longer smoking time entails a higher accumulation of PAHs, but a higher amount of fat seems to contribute even more to this accumulation in similar products [55]. In our experiment, the time factor is limited to only two days, but even so, in fatter duck meat, the rate shows uniformity in the accumulation of PAHs, regardless of the type of wood used in smoking.

Significant differences ($p < 0.001$) were recorded when the smoking time was the variable. The two-day smoking period dramatically increased the level of PAH contamination, especially in cherry wood.

ANOVA analysis based on wood type revealed a significant influence of the type of wood in PAH contamination, the most "polluting" being the cherry wood.

## 5. Conclusions

The highest concentrations of PAHs were present in duck meat, irrespective of smoking time (one vs. two days) or wood type (cherry, plum, and beech).

When wood type was interrogated, the higher concentration overall of PAHs was quantified when beech wood was used, followed by cherry and plum woods.

Two days of smoking contributed to higher concentrations of PAHs in meat, and the fastest PAHs accumulation was shown in duck meat, while turkey meat had the lowest total content of PAHs and also Σ4PAHs. The only meat that accumulated benzo(a)pyrene was turkey meat when cheery wood was used, but in low concentrations and far below the legally imposed limits.

As follow-up, the sensory analysis must complete the characterisation of poultry smoked breast. Moreover, lipid oxidation is a concern and should be approached through measuring the oxidative status of the meat and the likelihood of free radicals' occurrence after smoking.

**Author Contributions:** Conceptualization, C.O.C. and A.C.; methodology, C.O.C. and R.-M.R.-R.; software, C.O.C. and R.-M.R.-R.; validation, A.C., A.L., and A.B.; formal analysis, A.C., A.L., A.B., and O.M.; investigation, C.O.C., A.C., A.L., and O.M.; data curation, C.O.C. and R.-M.R.-R.; writing—original draft preparation, C.O.C. and A.C.; writing—review and editing, C.O.C. and R.-M.R.-R.; visualization, C.O.C., A.C., A.L., and O.M.; supervision, C.O.C. All authors have read and agreed to the published version of the manuscript.

**Funding:** This research received no external funding.

**Institutional Review Board Statement:** Not applicable. No animals were used for applying experimental factors on them. They were raised for meat production on a private farm and sold as eviscerated carcasses to farmers' market. Samples were taken from slaughtered refrigerated poultry, and the research did not interfere with poultry welfare and farming conditions throughout their raising period.

**Informed Consent Statement:** Not applicable.

**Data Availability Statement:** Data supporting reported results available, upon request, from the authors.

**Conflicts of Interest:** The authors declare no conflict of interest.

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
