# Peer review of "Polycyclic Aromatic Hydrocarbons (PAHs) Occurrence in Traditionally Smoked Chicken, Turkey and Duck Meat"

_agriculture, doi:10.3390/agriculture13010057_

Round 1

Reviewer 1 Report

Polycyclic Aromatic Hydrocarbons (PAHs) occurrence in Traditionally Smoked Chicken, Turkey and Duck Meat. Very interesting information were provided in the manuscript. However, the presentation of the results could be improved, it's difficult to follow sometimes. A factorial design could be more appropriate to perform the analysis. Using letters to separate the means also could help to understand the data. The result section contains only tables without any explanation of the numbers, this should be corrected. SEM is more appropriate in comparison with SD, I suggest changing to mean±SEM. 

Author Response

Honourable reviewer,

Thank you indeed for taking time in evaluating our manuscript and providing us useful suggestions in order to improve it.

We followed most of your suggestions, we shortened the tables with PAH concentrations by signaling the significance of differences using superscripts per row values and subscripts per column values. Also, we have replaced StDeviation with SEM values.

We decide to keep this manner of comparing results instead of providing a single signification for overall comparison of all 3 variables, because it is relevant to have them compared in a one by one model.

Thank you indeed and please let us know about any other improvements that we can add!

Please rest assured of our sincere consideration!

Reviewer 2 Report

The research in the manuscript describes an interesting topic, especially from the point of view of determining the qualitative properties of autochthonous poultry products.
The summary is a bit too short and there is not a single numerical data in it from which you could quickly see the obtained results.
In the manuscript, I would like to see the chemical composition of the meat, since the fat content is mentioned in the discussion, but it is nowhere in the results.
In addition, the tabular presentation of the results is unreviewable. Too much data are crammed into the table, which makes the display illegible. Try to simplify the table, especially so that the variables (type of wood and time) are clearly visible. It is also unnecessary to present the p-value precisely. Simplify it so that you have the usual three levels of significance.

Author Response

Honourable reviewer,

Thank you indeed for taking time in evaluating our manuscript and providing us useful suggestions in order to improve it.

We followed your suggestions, improving the abstract. However, we have to comply with the maximum length of 200 words.

We added the chemical composition of the three poultry meats in the Results sections as well the appropriate analytical methods in material and method section and in the references list, resynchronising, as well the citations throughout the body text .

We've reduced the tables with PAH concentrations by signaling the significance of differences using superscripts per row values and subscripts per column values. Therefore, we removed the p values. Also, we have replaced StDeviation with SEM values, on the suggestions of another reviewer that read our manuscript.

Thank you indeed and please let us know about any other improvements that we can add!

Please rest assured of our sincere consideration!

Reviewer 3 Report

comments for the authors 

The manuscript titled with " Polycyclic Aromatic Hydrocarbons (PAHs) occurrence in Traditionally Smoked Chicken, Turkey and Duck Meat  ". The manuscript talking about a good issue. Overall, the highest concentrations of PAHs were present in duck meat, irrespective of smoking time (one vs. two-days) or wood type (cherry, plum and beech).  When wood type was interrogated, the higher concentration overall of PAHs was quantified when beech wood has been used, followed by cherry and plum woods. Two-days smoking contributed to higher concentrations of PAHs in meat, and the fastest PAHs accumulation has been shown in duck meat, while turkey meat has the lowest total content of PAHs,.The manuscript written well. But it needs a minor revision.  The materials and methods should be shortening and also the conclusion should be concentrated in two or three sentences. Also, Authors should confirm analysis of the residual of PAHs and evaluation of validation method and determination of limet of detection and limit of quantification.

Author Response

Honourable reviewer,

Thank you indeed for taking time in evaluating our manuscript and providing us useful suggestions in order to improve it.

We have shortened the materials and methods whenever possible but we have to add a ne sub-section due to the necessity to introduce proximate composition analytical methodology.

In HPLC analysis method section we added info related to  the residual of PAHs and evaluation of validation method and determination of limit of detection and limit of quantification.

Also we have shortened the conclusions. We still prefer to keep the last conclusion when we suggest some follow-up of the research, as we were used to do in other manuscripts published with MDPI.

Thank you indeed and please let us know about any other improvements that we can add!

Please rest assured of our sincere consideration!